# Cardiac, respiratory, and gastric rhythms independently modulate motor corticospinal excitability in humans

Tahnée Engelen[1,2]*, Teresa Schuhmann[3], Alexander T. Sack[3], Catherine Tallon-Baudry[1]*

1 Cognitive and Computational Neuroscience Laboratory (LNC2), INSERM U960, Department of Cognitive Studies, Ecole Normale Supérieure, PSL University, Paris, France, 2 Department of Psychology, Faculty of Education and Psychology, University of Jyväskylä, Jyväskylä, Finland, 3 Section Brain Stimulation and Cognition, Department of Cognitive Neuroscience, Faculty of Psychology and Neuroscience, Maastricht University, Maastricht, the Netherlands

* tahnee.engelen@yahoo.nl (TE); catherine.tallon-baudry@ens.psl.eu (CT-B)

## Abstract

Interoception refers to the brain's sensing of internal body state and encompasses various bodily systems, notably the cardiac, respiratory, and gastric rhythms. Beyond their roles in physiological regulation and emotional states, each of these visceral rhythms has been shown to influence brain activity and cognition, prompting for the development of various interpretative functional frameworks. However, both experimental data and functional hypothesis leave it unclear whether and how each visceral rhythm acts simultaneously and independently on brain activity. Here, we address this question by measuring in human participants how the corticospinal excitability of the motor system varies with the phase of each of the three visceral rhythms. We applied single pulse transcranial magnetic stimulation (TMS) over the hand region in primary motor cortex to elicit Motor Evoked Potentials (MEPs), whose amplitude reflects corticospinal excitability, and tested whether MEP amplitude depends on the phase of the simultaneously measured cardiac, respiratory, and gastric rhythms. All three visceral rhythms were coupled to motor excitability with similar effect sizes at the group level. However, we found no relation between coupling strengths: participants displaying high coupling with one organ did not necessarily display high coupling to the other organs. These results indicate that independent mechanisms could underly the coupling between the cardiac, respiratory, and gastric rhythms and motor excitability. We further introduce the concept of individual interoceptive profiles and show that such interoceptive profiles obtained from objective coupling strength measures were not explained by self-reported awareness of the organ. Altogether, our results call for refined specifications of the frameworks offering a functional or clinical interpretation of viscera–brain coupling taking into account both independent mechanisms and individual interoceptive profiles.

**Data availability statement:** The preprocessed data used for the analyses reported in this article are available through https://osf.io/6utgf/ and summary data to replicate figures 2 and 3 are available as S1 Data and S2 Data. Custom-written code for the analyses are available at https://doi.org/10.5281/zenodo.17307040. To comply with EU laws, individual data can only be shared once an institutional data sharing agreement with Maastricht University has been signed. Due to data protection regulations in the EU, individual anatomical MRI data cannot be made available online.

**Funding:** This work was funded by Agence Nationale pour la Recherche (https://anr.fr, ANR- 17-EURE-0017 and ANR-10- IDEX-0001-02 awarded to PSL University, and ANR- 21-CE37-0031 awarded to CTB). CTB is supported by a senior fellowship of the Canadian Institute For Advanced Research (CIFAR, https://cifar.ca) program in Brain, Mind, and Consciousness. The funders had no role in study design, data collection and analysis, decision to publish, or preparation of the manuscript.

**Competing interests:** The authors have declared that no competing interests exist.

**Abbreviations:** BF, Bayes Factor; BMI, body mass index; ECG, electrocardiogram; EGG, electrogastrogram; EMG, electromyography; FDI, first dorsal interosseous; FDR, False Discovery Rate; GLM, general linear model; MAIA-2, Multidimensional Assessment of Interoceptive Awareness, Version 2; MEPs, Motor Evoked Potentials; THISQ, Three-domain Interoceptive Sensations Questionnaire; TMS, transcranial magnetic stimulation.

## 1. Introduction

Interoception refers to the sensing of internal bodily signals, for which signals from the heart, lungs, and stomach are considered core components, both in the original definition by Sherrington [1] and in more recent views [2,3]. The function of interoception is obviously associated with homeostatic regulations, and as such is involved in responses to physiological challenges or emotional contexts. However, experimental evidence suggests a functional role for interoception beyond physiological regulation. In perceptual and cognitive tasks that have no apparent link to homeostatic constraints and are emotionally neutral, a growing number of experimental studies find coupling between a visceral rhythm and brain activity and/or reaction times [4,5]. Such findings have motivated a number of theoretical proposals hypothesizing on the functional role of the interaction between interoception and the cognitive brain and behavior.

Some proposals focus specifically on action, such as active sensing, i.e., the notion that sensing external information is an active process involving motor sampling [6]. Respiration plays a key active sensing role for olfaction, because inhalation is required for odor detection [7]. By extension, experimental findings involving other sensory modalities and other visceral rhythms have been interpreted in the active sensing framework, for instance, the over-abundance of saccades to explore the visual environment in the early phase of the cardiac cycle [8,9]. Other functional proposals are even more general. For instance, visceral rhythms could provide a temporal scaffold to facilitate large-scale neuronal coordination, with experiments on the coupling of respiration and limbic regions in rodents [10,11] or on the coupling between the slow gastric rhythms and all sensory and motor cortices in humans [12,13] interpreted along those lines. Both lines of thinking are appealing and not necessarily mutually exclusive. The point we want to raise here is that such frameworks are applied relatively indiscriminately to any visceral rhythm. In other words, there is an unformulated underlying assumption that although the cardiac, respiratory, and gastric rhythms support distinct physiological functions and operate at markedly distinct dominant frequencies (in humans, respectively, 1, 0.2, and 0.05 Hz) (for review, see [5]), they can nevertheless fit within the same interpretative framework. Indeed, it is possible that visceral rhythms exert an influence in a co-dependent manner, for example, through global changes in arousal states mediated through the vagus nerve [14]. Still, this underlying assumption remains to be empirically tested. In the field of interoception, even if multiple organs are considered they are typically limited to cardiac and respiratory signals [15–19]. When it comes to transcranial magnetic stimulation (TMS), only the cardiac rhythm has been measured so far [20–24]. To date, no experiment measured all three visceral rhythms, leaving their potential independent influences on the brain speculative.

Here, as a first attempt to bridge the gap between generic interpretative frameworks and rhythm-specific experimental data, we ask whether the excitability of the motor system is simultaneously and independently modulated by the cardiac, respiratory, and gastric rhythms. The motor system is a good candidate since movement onsets [8,9,25–27], response inhibition [28,29], discharge of alpha motoneurons [30],

and motor cortico-spinal excitability [20] vary along the cardiac cycle. The phase of respiration affects the timing of voluntary [16,31] and involuntary [32] actions, motor-related neural activity [16,33], and changes in breathing style influence motor cortico-spinal excitability [34,35]. Finally, the gastric rhythm is consistently coupled to motor and premotor areas at rest in humans [12,13] as well as in anaesthetized rodents [36,37]. In practice (Fig 1), we probed non-invasively cortico-spinal excitability in human male and female participants by measuring from a hand muscle the Motor Evoked Potentials (MEPs) induced by TMS over primary motor cortex [38,39]. MEPs typically measure excitability from primary motor cortex down to the final muscle where they are measured, including spinal relays [39]. We tested for relationships between MEPs and the phase of the simultaneously recorded cardiac, respiratory, and gastric rhythms.

## 2. Materials and methods

### 2.1. Ethics statement

Twenty-nine healthy participants completed the experiment after providing written informed consent. The study was approved by the Ethics Review Committee Psychology and Neuroscience of Maastricht University (reference number ERCPN-247_01_01_2022). The study was conducted according to the principles expressed in the Declaration of Helsinki.

### 2.2. Participants

Participants received a voucher reward of 10 euros per hour. Inclusion criteria were to be aged above 18 years and to have a body mass index (BMI) lower than 25 to ensure good quality gastric recordings [40]. Participants were further screened for MRI and TMS safety. One participant was excluded from analysis due to issues with recording of the respiration signal. In total, 28 participants were included in the final analysis (8 male, 2 left-handed, mean age(SD) = 27.36(8.98),

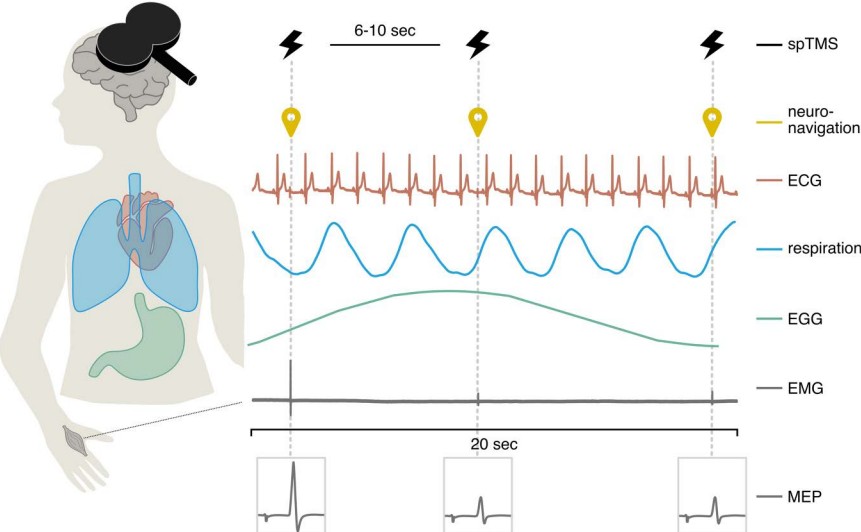

**Fig 1. Experimental set up.** Single pulses of TMS were applied to the hand area of the right primary motor cortex with an inter-pulse interval randomized between 6 and 10 s. Simultaneously, the neuronavigated coil position (yellow), electrocardiogram (red), respiratory signal (blue), electrogastrogram (green), and electromyography from the left hand (gray) were recorded. The figure shows traces of a 20-s time segment from one participant of the cardiac (raw), respiratory (filtered), gastric (filtered), and EMG (raw) signals. The experimental measure was the Motor Evoked Potential amplitude measured on a hand muscle (first dorsal interosseous), analyzed against the phase of the cardiac, respiratory, and gastric rhythms. Note that the three rhythms have very different periods (~1 s for the heart, ~5 s for respiration, and ~20 s for the gastric rhythm). Abbreviations: spTMS, single pulse transcranial magnetic stimulation; ECG, electrocardiogram; EGG, electrogastrogram; EMG, electromyogram; MEP, motor evoked potential.

mean BMI(SD) = 21.76(2.38)). It was verified that the two left-handed participants were not outliers, i.e., their coupling scores with each of the viscera did not deviate more than three standard deviations from the group mean.

## 2.3. Experimental procedure, general organization

The experiment consisted of two sessions. During the first session, an anatomical MRI scan was acquired at rest. At the start of the second session, participants completed the questionnaires, after which all physiological recordings were prepared and TMS intensity and stimulation site were determined. Participants completed three blocks of the task, each lasting around 16 min, while the TMS stimulation was applied. Participants were in a seated position with their head supported by a chin rest, were instructed to fixate on a screen, and were given a task to avoid drowsiness (see Section 2.6. for task details).

## 2.4. TMS application and neuronavigation recordings

Single pulse TMS was applied over right M1 hotspot using a MC-B70 figure-of-eight coil and Magpro X100 stimulator (MagVenture A/S, Farum, Denmark). Participants were co-registered to their individual anatomical T1 MRI (TR = 2,300 ms, TE = 2.98 ms, TI = 900 ms, 192 slices, FOV = 256 mm, matrix size = 256 × 256, 1 mm isotropic voxel, acquired on 3T Siemens Prisma Fit) and coil position was recorded throughout the session using a neuronavigation system (Localite GmbH, Bonn, Germany). The coil was placed tangentially to the scalp with the handle pointing backward at a 45° angle from the midline. Individual stimulation intensity was set to elicit an MEP of approximately 1 mV in amplitude (mean stimulation intensity(SD) = 41((7)% maximum stimulator output). Motor threshold was not used to determine stimulation intensity. The stimulation target was identified by applying single pulses over the right hemisphere until the location showing a maximum response from the left first dorsal interosseous (FDI) muscle was found, which was set as the target for the rest of the session. The interpulse interval was jittered between 6 and 10 s and a total of 360 pulses were applied during the task, subdivided into three blocks.

The difference between intended and real target location was calculated off-line as follows:

$$\text{Distance-to-target} = \sqrt{(dP^2 + dS^2 + dL^2)}$$

where $dP$, $dS$, and $dL$ are the difference between current target and the predefined target in the Posterior, Superior, and Lateral dimensions. Trials on which the distance-to-target exceeded more than four standard deviations from the mean of the participant were excluded from analysis. Mean(SD) distance-to-target of included trials was 3.07(1.7)mm.

## 2.5. Physiological recordings

Physiological signals were recorded using a BrainAmp ExG amplifier and the BrainVision Recorder software (Brain Products GmbH, Gilching, Germany) at a 5,000 Hz sampling rate. All signals were recorded with a high cut-off of 1,000 Hz and with a low cut-off filter set to a time constant of 10 s for electromyography (EMG) and the electrocardiogram (ECG) and DC for respiration and the electrogastrogram (EGG). A respiration belt containing a pneumatic pressure-sensitive cushion was placed around the chest to record thoracic respiratory movements. All other signals were recorded with bipolar montage using cutaneous AgCl ring electrodes. Motor evoked responses were recorded using EMG of the left FDI muscle following the belly-tendon montage with one electrode placed over the belly of the muscle and a reference electrode placed along the outside of the index finger. The ECG was recorded by placing one electrode below the right clavicle and the other electrode on the left lower abdomen (lead II). The EGG was recorded using a grid of six electrodes on the left side of the belly, all referenced to an electrode on the upper right side of the belly, following recommendations of Wolpert and colleagues (2020).

## 2.6. Task and eyetracking

To help participants remaining wakeful during data acquisition, one quadrant of the black circle surrounding the fixation point could occasionally turn white for 500ms. In each block 16 targets were randomly presented, always starting/ending with at least 500ms distance to the TMS pulse. To avoid contamination of manual motor actions on MEP amplitude, participants responded by making an eye movement to the corner of the screen in the direction at which the target appeared. Eye position was tracked using an Eyelink 1000 system (SR Research, Canada), using a monocular recording of the right eye with a 16mm lens at a sampling rate of 1,000 Hz.

## 2.7. Questionnaires

To measure self-reported interoception, participants completed two questionnaires prior to the start of the experiment, the Three-domain Interoceptive Sensations Questionnaire (THISQ) [41] and the Multidimensional Assessment of Interoceptive Awareness, Version 2 (MAIA-2) [42].

## 2.8. Data pre-processing

Offline pre-processing and analysis of physiological signals was done using the Fieldtrip toolbox implemented in Matlab (Matlab version 2024a and Fieldtrip version 20220104) [43] and additional custom-built Matlab code. Continuous EMG data was band-pass filtered between 1 and 1,000 Hz using a 4th order forward and backward Butterworth filter and segmented around the TMS pulse (−1 to +1s). MEP amplitude was calculated as the difference between the local maximum and local minimum (peak-to-peak) between 20 and 60ms after the pulse, falling well within the range in which MEP peaks typically occur [44,45]. Following standard practice [46] and in line with previous MEP experiments (e.g., [20,47]), MEPs were excluded if their amplitude was below 50µV. MEPs were furthermore excluded if their amplitude got cut off due to amplifier saturation, or if the signal to noise ratio (SNR) between the time preceding the pulse relative to the MEP amplitude was below three. SNR was estimated by dividing the raw peak-to-peak MEP amplitudes by the maximum Root Mean Square value in the 500−10ms preceding the pulse. For the first 11 participants, the resolution of the EMG channel was set to 0.1µV per bit, leading to occasional amplifier saturation. In those cases, MEPs were excluded. For the remaining participants, the resolution was set to 0.5µV per bit. For all reported analyses, selected peak-to-peak MEP amplitudes were z-scored (see Section 2.8.2).

Continuous ECG data was band-pass filtered between 1 and 40 Hz (windowed-sinc FIR filter, one pass-zero phase shift). To identify the PQRST waves on the ECG, we followed the procedure described in [48]. Briefly, for each participant and block, we establish a template of the cardiac cycle (by averaging a few prototypic cycles) and correlate the template with the continuous ECG time series. R-peaks were identified on the normalized correlation as exceeding a threshold of 0.6. Misidentified R-peaks were manually corrected. Further cardiac events (P, Q, S, and T waves) were defined relative to the identified R-peak as local minima or maxima in preset time windows. In particular, the peak of the T-wave was identified as the local maximum occurring within 420ms after the Q-wave.

We define systole as the time between R-peak and end of T-wave, and diastole as end of T-wave until the next R-peak (see Fig 3A).

Noisy cycles (i.e., with an unclear T-wave) were identified by excluding any cycles in which the RT interval exceeded three standard deviations from the mean of that block. For the main analysis, instantaneous cardiac phase was calculated by assigning radian values between 0 and 2π to each TMS pulse based on its relative timing to the R-peak preceding the pulse and the first R-peak following the pulse.

To identify the end of the T-wave, we used the second derivative of the signal: a participant-specific block-by-block mean T-wave was computed, and the local maximum in the second derivative of the mean T-wave between the peak of T up to +150ms was taken as end of the T-wave. Given the known low variability of the QT interval under resting conditions

(standard deviation <5 ms, [49]), a fixed duration of cardiac systole was calculated for each participant by taking the mean end of T time of the three blocks (i.e., mean R to end of T duration). Cardiac diastole was then defined as the remained of the cardiac cycle (i.e., end of T to the next R peak).

Continuous respiration data was smoothed using a 750 ms window moving average (Matlab movmean). Expiration onset was identified on the *z*-scored respiration signal using the findpeaks function with a minimum peak distance of 1 s and minimum peak prominence set to the mean positive peak height excluding extreme values (peaks exceeding three times the standard deviations from the mean signal are excluded). The beginning of inspiration was determined by using an adapted version of the trapezoid area algorithm [50]. Briefly, this involved iteratively fitting trapeziums on the derivative of the respiration signal, and finding the area where the trapezoid has the largest area, which is taken as inspiration onset. Artefacted segments were identified by manual inspection. Cycles with either inspiration or expiration length exceeding the mean length of inspiration and expiration of the respective block by four SDs were excluded from further analysis. Instantaneous respiratory phase was calculated by assigning radian values between 0 and $2\pi$ to each TMS pulse based on its relative timing to the start of inspiration and end of expiration interval.

EGG data was preprocessed following the steps described in Wolpert and colleagues (2020). Data was demeaned, detrended, and down-sampled to 1,000 Hz. A Fast-Fourier Transform with a Hanning taper was used to identify the channel with the largest spectral activity in the normogastric range (0.033–0.067 Hz), which was then selected for further analysis. Data was bandwidth filtered around individual peak frequency ±0.015 Hz using a third-order finite impulse responses filter (Matlab FIR2). Artefacted segments were identified as cycles whose length exceeded three SDs from the mean cycle length (except two participants where SD criterion of two was used due to noisier data) or which displayed nonmonotonic changes in phase. Instantaneous phase was retrieved by Hilbert transforming the filtered data. After preprocessing, the mean(SD) length of clean gastric cycles was 20.5(1.3) s, and mean standard deviation (SD) of participant's gastric cycle was 1.95 (min SD: 0.99; max SD: 3.34).

After preprocessing, a mean(SD) of 287(41) out of 360 trials per participant remained for analysis (min: 191; max: 342). Per participant, a mean(SD) number of trials were excluded due to either MEP amplitude or pre-MEP noise in the EMG (24.6(27.5)), distance-to-target (1.3(1)), cardiac signal (2.5(4.6)), respiratory signal (36.6(37.3=), or gastric signal (14.3(9.8)) (note that a trial was excluded if it failed inclusion criteria for one of the signals, but could be labeled for exclusion due to multiple signals showing issues at once).

## 2.9. Statistical analysis

**2.9.1. General.** General support for or against the null hypothesis was quantified using the Bayes Factor (BF) using Matlab code provided by Liang and colleagues (2008) [51] and based on the models of [52,53]. $BF_{10}$ values smaller than 0.33 indicate moderate support for $H_0$, and values smaller than 0.1 indicate strong evidence for $H_0$. On the other end, values exceeding 3 indicate support for $H_1$, and values exceeding 10 indicate strong evidence for $H_1$. Reported effect sizes are quantified using Hedge's g (Matlab meanEffectSize), where effects above 0.2 are considered small, 0.5 considered medium, and 0.8 considered large. Reported correlations are calculated based on a ranked correlation test (Kendall's tau B implemented in JASP (JASP Team (2024))) and a False Discovery Rate (FDR) correction of *p*-values is applied if necessary. The pre-specified alpha for all reported statistics is set to 0.05.

**2.9.2. The effect of TMS coil distance-to-target on MEP amplitude.** The distance of the TMS coil to the target stimulation site in the motor cortex is known to influence MEP amplitude and previous work has suggested controlling for such displacements increases statistical sensitivity to detect effects [54,55]. This distance varies with experimenter and/or participant movement and might thus be in particular related to the participant's breathing movement, which would confound all further analysis. Here, to reduce confounding effects of coil displacements and increase statistical power, we account for the effect of displacement of the TMS coil with respect to the participant's head. By modeling the influence of distance-to-target on the MEP amplitude, we in particular correct for the effect of head displacement caused by the

 

PLOS Biology

participant's respiratory rhythm contributing to phase-locked amplitude changes. Note that this conservative approach is also likely to remove any physiological component related to breathing-related movement. We modeled the trial-by-trial *z*-scored peak-to-peak amplitude of the MEP with the *z*-scored distance-to-target value for each participant separately using a general linear model (GLM) in the following manner:

GLM Model 1:

$$[z(\text{MEP}_{\text{amplitude}})] \sim \beta_0 + \beta_{\text{distance-to-target}} * [z(\text{distance-to-target})] + \text{MEP}_{\text{residuals}}$$

To test for significance, the beta values for the distance-to-target regressor were tested against zero using a two-tailed paired *t* test. Given the significant result of this analysis, all subsequently described analyses are performed on the residuals resulting from this model. In other words, our analyses of interest are performed on amplitude values which have been corrected for the effect of distance-to-target.

**2.9.3. Assessing phase–amplitude coupling between visceral rhythms and MEPs.** To test whether the amplitude of the MEP was coupled to the phase of either the cardiac, respiratory, or gastric rhythm (visceral phase with MEP amplitude coupling), we used a circular spline generalized circular model [56]. This method allows for computing participant-specific chance levels of phase–amplitude coupling, meaning it does not require phase-alignment on the group level, which is unlikely to be the case for the gastric rhythm where there is no true zero point in the cycle [40]. Briefly, the circular GLM (normal distribution, identity link function) involves creating two models, the spline model in which cardinal splines are fitted to each participant's data to estimate amplitude as a function of phase, and a null model. The spline model is fitted based on a number of control points which is estimated for each participant separately using the Akaike information criterion (mean(SD) control points for cardiac = 4.5(1.4), for respiration = 4.7(1.3), and gastric = 4.7(1.6)). The null model consists of data points whose values are set to equal the mean of the participant's data, along the phase of the visceral rhythm, thereby creating a uniform distribution (where amplitude does not change as a function of phase). To quantify phase–amplitude coupling, the maximum absolute deviation between the spline and null model is calculated (*r*-value). The larger the *r*-value, the more the empirical data deviate from a uniform distribution. Given that the largest deviation can occur at any phase within a visceral cycle, and *r*-values are computed per participant, this model thus does not require phase-alignment of coupling effects on the group level. Rather, we here tested whether, in general MEP, amplitudes are coupled to visceral phase higher than chance level (see below), irrespective of the phase in which each participant showed this coupling. We ran one model *per visceral signal* (thus three models) and *per participant* to obtain individual empirical *r*-values.

Importantly, large *r*-values could be obtained by chance. To estimate the chance level, we adopted the same approach previously used to demonstrate coupling between the phase of the gastric rhythm and the amplitude of neural alpha oscillations [57]. In practice, for each participant, we created chance-level coupling by randomly shuffling the link between (cardiac, respiratory, or gastric) phases and MEP amplitudes, ran the model described above, and compute the *r*-value for this specific permutation of the data. We repeated this procedure to obtain 10,000 *r*-values obtained by chance. The participant's coupling strength was then computed by subtracting the median of the permuted *r*-value distribution from the empirical *r*-value. We then tested whether at the group level, coupling strengths were significantly larger than zero using a one-sided Wilcoxon signed-rank test against zero. Here, we test whether empirical coupling was higher than permuted chance level, and thus one-sided tests were used. Of note, while coupling strength might occasionally be negative (but small) in a given participant, at the group level two signals cannot be significantly less phase-amplitude coupled than what would be expected by chance. Testing the other side of the hypothesis (empirical coupling lower than chance) would thus not be justified.

Because circular spline GLMs allow for only one circular regressor, we thus tested each visceral rhythm separately. Differences in SNR between the different visceral signals (i.e., sensitivity to accurately detect the phase of each rhythm at

the time of the pulse) might impact the accuracy of the spline model. However, potential differences in the quality of phase estimation are also taken into account when estimating chance level per participant per visceral rhythm. The final index we used, coupling strength, is thus relatively immune to SNR and phase extraction issues. To correct for the effect of running one test per visceral rhythm, and this potentially inflating $p$-values, we applied an FDR correction to the obtained $p$-values.

Lastly, we wanted to test for the inter- or independence of coupling of the different visceral rhythms to the motor system in another manner besides correlations between the coupling strengths, to ensure our findings are not simply a result of our chosen analysis technique. To this end, we opted for a simple GLM, similar to the one used to estimate the effect of distance-to-target on MEP amplitude, in which all three visceral rhythms are entered as well as their interactions. To be able to analyze circular data, such as the visceral phase, in a simple GLM required the binarization of the data. To this end, the visceral phases were split into one "excitatory" and one "inhibitory" phase based on each individual's data. As these windows of excitation and inhibition were based on individual data, they could thus differ between participants and were not necessarily linked to specific physiologically meaningful phases. The excitatory and inhibitory phases were established for each participant and each visceral rhythm as follows: data was divided into six equally spaced phase bins between 0 and 2pi. Excitatory phase was defined as the phase bin with the highest mean residual MEP values and the two adjacent bins, and inhibitory phase consisted of the three remaining bins.

This approach allowed us to add all three visceral rhythms and their interactions into the same model (rather than one model per signal), which was not possible in the circular spline model. The influence of binary visceral phase on the residual values was then estimated in the following way:

GLM Model 2:

$$MEP_{residuals} \sim \beta_0 + \beta_{cardiac\ phase} + \beta_{respiratory\ phase} + \beta_{gastric\ phase} + \beta_{cardiac\ *\ respiratory\ phase} + \beta_{respiratory\ *\ gastric\ phase} + \beta_{cardiac\ *\ gastric\ phase} + \beta_{cardiac\ *\ respiratory\ *\ gastric\ phase}$$

Importantly, this model did not aim to see if the phase of each visceral rhythm explains the MEP residuals, as this was already established with the spline models, and phase effects are here enforced by splitting the data based on optimal phase. Rather, the goal was to be able to investigate the interaction effects on the MEP residuals, as reveal whether the visceral phase effects previously observed are independent. To test for significance of the interaction terms, the beta values of the interaction term regressors were tested against zero using a two-tailed paired $t$ test.

Lastly, to quantify how flat (i.e., homogeneous profile with similar levels of coupling with all three visceral rhythms) or peaky (i.e., heterogeneous profile with, for instance, large coupling for one organ and low coupling for the other two), we computed the following variability index (vi) for each participant:

$$vi = abs(CS_{cardiac}-meanCS) + abs(CS_{resp}-meanCS) + abs(CS_{gastric}-meanCS)$$

where meanCS is the mean of $CS_{cardiac}$, $CS_{resp}$, and $CS_{gastric}$. Here, larger values indicate more heterogeneous interoceptive profiles and smaller values more homogeneous profiles. This variability index of all participant's interoceptive profiles is reported in S1 Fig.

**2.9.4. The relationship between visceral–motor coupling and self-reported interoception.** We wanted to test for relationships between objective measures of brain–viscera coupling and the extent to which participants themselves indicate to be aware of their internal organs. To do so, we correlated each participant's organ-specific coupling strength with the matching organ–specific scores obtained in the THISQ questionnaire (e.g., cardiac coupling strength correlated with cardiac THISQ score). We also tested for a correlation between coupling strengths and a widely adopted interoception questionnaire, the MAIA-2 [42]. For the correlation analysis we focused on the correlations between the coupling strengths and the body listening and noticing sub-scores, which refer to the awareness of comfortable,

uncomfortable, and neutral bodily signals and active listening to the body for insight, respectively, as they were closest to our research questions. The other dimension of the MAIA (Not Distracting, Not Worrying, Attention Regulation, Self-Regulation, and Trusting) were not correlated with coupling scores, as these dimensions mostly relate to attention to pain and discomfort of the body, and the ability to change one's attention to bodily signals [58], which was not relevant to the current experiment. Correlations with the MAIA questionnaires were FDR corrected for six tests (two sub-scores by three coupling strengths).

**2.9.5. Inter-individual variability in the alignment to physiological phases.** The circular spline GLM specifically tests whether there is phase–amplitude coupling between the visceral rhythms and corticospinal excitability, however, it does not conclude anything about whether the exact optimal phase is consistent between participants. This means that one participant could have a strong increase in excitability early in the cardiac phase and another late in the cardiac phase. However, based on existing literature, such increases in motor excitability might occur in specific moments in either the cardiac or respiratory cycle (e.g., Al and colleagues, 2023; Park and colleagues, 2020). To test for this possibility, we ran an exploratory analysis in which the data was divided into six equally spaced phase bins between 0 and 2pi for each participant, and extracted their mean bin residual value, and which bin had the highest mean. We here specifically focused on the data wrapped to the physiologically meaningful phases, so bins between zero and pi represent either systole or inspiration, and between pi and 2pi represent either diastole or expiration. This analysis allows looking at whether group mean bin values show patterns across the cycle, and whether the highest bin is consistent on the group level. Note that here we do not further examine the gastric rhythm, as the exact starting point of each cycle is unknown, and thus looking at group level consistency is not meaningful.

## 3. Results

### 3.1. Distance-to-target explains MEP amplitude

Our first step was to quantify the influence of distance-to-target on MEP amplitude (GLM model 1). Because there might be small shifts in the relative position of the coil and the participant's head, notably with breathing, the coil might stimulate the cortex at a location slightly different from the one intended. As expected from the literature [54,55], a smaller distance between intended and real TMS target location resulted in larger MEP amplitudes (paired -$t$ test against 0, $t(27) = -5.26$, $p < 0.001$), with a $BF_{10}$ of 1,475 indicating strong support for the alternative hypothesis, and a Hedge's g of $-0.97$ indicating a large effect (see Fig 2A). All subsequent analyses are performed on MEP fluctuations unrelated to distance-to-target fluctuations, i.e., the residuals of this first model.

### 3.2. The amplitude of MEP varies with the phases of the three visceral rhythms

We first wanted to establish whether, as hypothesized, there was coupling between the phase of each of the visceral rhythms and the amplitude of the MEP. Phases are defined from R peak to R peak for the cardiac rhythm, from beginning of inspiration to end of expiration for the respiratory rhythm, and are unrelated to specific physiological events for the gastric rhythm, where such markers do not exist [40].

Using a circular spline GLM approach [56], we estimated, for each participant, how much MEP amplitude varies with the phase of each visceral rhythm. We computed coupling strengths, reflecting the difference between empirically obtained coupling scores and the median of chance-level coupling scores obtained by permutation. At the participant level, a large positive coupling strength value indicates more coupling than expected by chance, while a small positive or negative coupling strength indicates chance-level coupling. At the group level, we found coupling strengths to be significantly larger than zero for the cardiac (one-tailed Wilcoxon signed-rank test against zero, $z = 1,79$, $p_{FDR} = 0.047$, $BF_{10} = 3.84$, Hedge's $g = 0.55$), respiratory ($z = 2.65$, $p_{FDR} = 0.012$, $BF_{10} = 22.86$, Hedge's $g = 0.54$), and gastric ($z = 1.67$, $p_{FDR} = 0.047$, $BF_{10} = 2.87$, Hedge's $g = 0.50$) rhythms (see Fig 2B). This result demonstrates that fluctuations of corticospinal excitability over time are driven by the phase of each of the three visceral rhythms, with similar effect sizes at the group level.

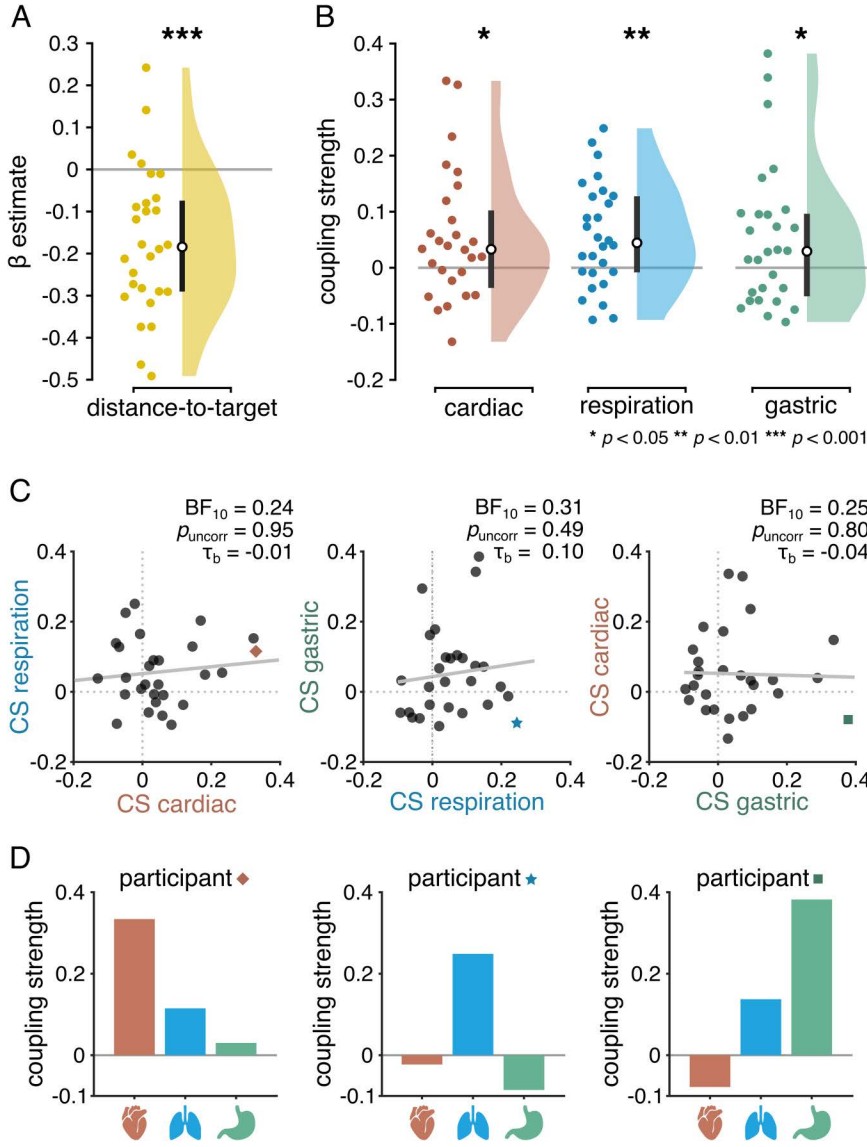

**Fig 2. Sources of MEP amplitude fluctuations. A)** The distance between the TMS coil and the cortical target in motor cortex significantly affect MEPs, with larger distances resulting in smaller MEPs. In all subsequent analyses, the influence of distance-to-target is regressed out from MEP amplitude. **B)** Coupling strengths (CS) between each visceral rhythm and MEPs. CS reflects the difference between empirical and chance level coupling estimated with a circular GLM. Coupling strengths were significantly larger than zero for the cardiac, respiratory, and gastric rhythms. This means that the phase of the visceral rhythms impacted the amplitude of the MEP. **C)** Correlations between coupling strengths (CS) for each possible pair of visceral rhythms. There was moderate evidence for an absence of correlation between the coupling strengths of the different organs, showing that the phase–amplitude coupling between the viscera and motor system happened independent from one another. Note that for illustration purposes a regression line has been added to the figure, whereas the reported statistics are the result of a ranked correlation test (Kendall's tau). Notably, in the left panel, the slope of the regression line is positive whereas the tau value is negative. **D)** Each participant displayed a unique interoceptive profile. To illustrate this inter-individual variability, interoceptive profile of the participant with the largest cardiac–motor coupling (left, red lozenge in C), the participant with the largest respiration–motor coupling (middle, blue star in C), and the participant with the largest gastric–motor coupling (right, green square in C). As can be seen from the bar plots depicting participant's interoceptive profiles, even though each of these three participants had the highest coupling scores between corticospinal excitability and one of the visceral rhythms, coupling with the two other visceral rhythms was not necessarily large. More profiles are presented in S1 Fig, and data supporting panels A, B, and D are available in S1 Data. * $p < 0.05$; ** $p < 0.01$; *** $p < 0.001$.

To ensure the circular model on respiratory phase was not biased by the distance-to-target correction (i.e., running the model on the MEP residuals), we re-ran this model using the $z$-scored raw MEP values, and obtained similar results ($z = 1.92$, $p = 0.027$, $BF_{10} = 3.02$, Hedge's $g = 0.49$).

### 3.3. Cardiac, respiratory, and gastric phase influences on corticospinal excitability vary independently from each other

At the participant's level, do we find evidence for interoceptive-sensitive or interoceptive-insensitive participants? In other words, is high coupling with one visceral rhythm predictive of high coupling with another visceral rhythm? We used two different approaches to test for potential independence. Firstly, we tested for correlations between the individual coupling strengths of the three visceral rhythms with each other (Fig 2C). We found no correlation between coupling strengths, with all tested correlations providing moderate evidence for the null hypothesis (Kendall's tau B correlations, all $p$ values > 0.49, all $BF_{10} < 0.33$). By exploring the interoceptive profiles of the participants with the highest coupling strength of each rhythm, it is apparent that high coupling with one rhythm does not necessarily mean high or even any coupling with another rhythm, and all types of interoceptive coupling profiles can be possible, as illustrated in Fig 2D and see S1 Fig.

We confirmed those results using a second approach, where all rhythms were combined in the same GLM. To switch from a circular spline model to a GLM with interactions between rhythms, we defined for each participant and each rhythm an interval of pi where MEP amplitude was maximal (excitatory phase), and an interval of pi where MEP amplitude was minimal (inhibitory phase). For each participant, we then accounted for MEP amplitude by binarized phases for all three rhythms, and, importantly, all possible interactions between rhythms. The model confirms a significant influence of all three rhythms, as expected by the model design, but revealed an absence of interactions between rhythms: We found no double or triple interactions between any of the visceral phases (all $p$-values for interaction terms > 0.18). Except for the interaction term between cardiac and gastric phases ($BF_{10} = 0.47$), all $BF_{10}$ values indicate moderate evidence for the null hypothesis for all other interactions ($BF_{10} < 0.33$).

Together, these two separate analyses converge on the same conclusion, which is that even though the phase of each of the three visceral rhythms influences the amplitude of the MEP, they do so in an independent manner: at the participant's level, high coupling between motor excitability and one rhythm provides no information of the coupling with another rhythm, resulting in a large variety of individual interoceptive coupling profiles.

### 3.4. Alignment to cardiac and respiratory phase shows inter-individual variability

The cardiac and respiratory rhythms can be subdivided in two meaningful physiological phases, respectively, systole/diastole (corresponding to cardiac contraction and relaxation) and inspiration/expiration. In an exploratory analysis, we find that, in line with previous findings [20], at the group level, corticospinal excitability was enhanced in the systolic cardiac phase (Fig 3A, left). In terms of variability, 20 out of 28 participants had their highest phase bin in the systolic phase (Fig 3A, right), showing that most, but not all, participants followed the pattern for cardiac enhancement during systole. Regarding respiration, at the group level, corticospinal excitability was largest in early inspiration (Fig 3B, left). This pattern is consistent with previously reported increased amplitudes of the readiness potential during inspiration [16], although the same study reported more action onsets during the end of expiration. Inter-subject variability appeared a bit larger than for cardiac phase: 17 out of 28 participants had their maximal MEP amplitude during inspiration (Fig 3B, right).

While group data suggest preferential phases of enhanced excitability, during cardiac systole and at the beginning of inspiration, there was also large inter-subject variability. We therefore further explored whether participants who deviated from the group pattern for cardiac phase also deviated from the group pattern for respiratory phase. As illustrated in Fig 3C, some participants followed the group pattern for both systole and inspiration (participant 8), others showed systolic coupling, but late expiration coupling (participant 7), some showed no coupling to the cardiac cycle, but an increased

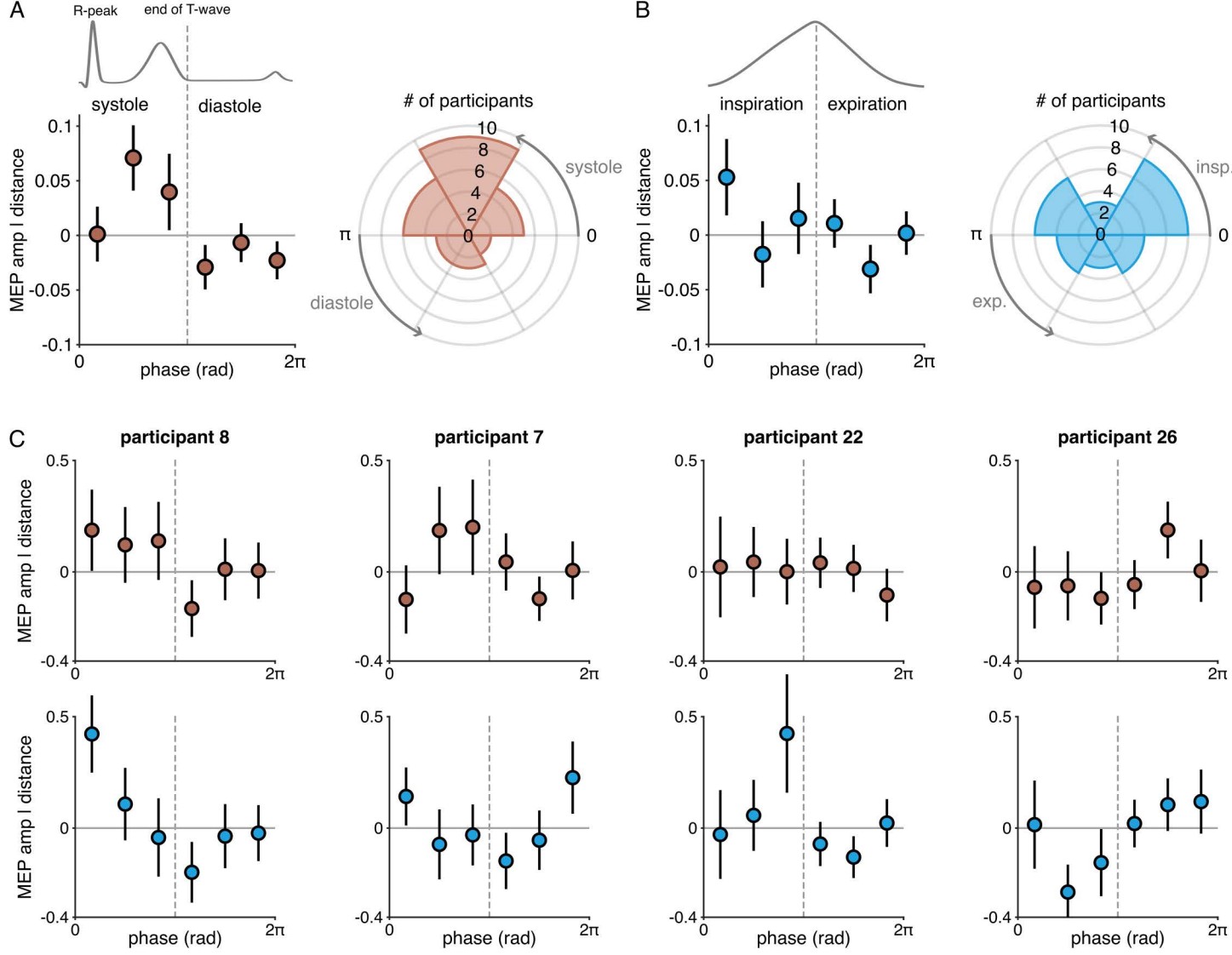

**Fig 3. Exploratory analyses on alignment to cardiac systole/diastole (corresponding to cardiac contraction and relaxation) and inspiration/expiration. A)** Mean MEP amplitudes after correction for distance-to-target effects (MEP amp | distance) binned into six phases, reflecting either systole (bins 1–3 with bins spaced equidistantly between 0 and pi) and diastole (bins 4–6 with bins spaced equidistantly between pi and 2pi). The polar histogram (right) indicates the number of participants having their maximal coupling strengths in the bin. In 20 out of 28 participants, cardiac-motor excitability coupling peaked in systole. **B)** Mean MEP amplitudes corrected for distance-to-target effects (MEP amp | distance) binned into six phases, reflecting either inspiration (bins 1–3 with bins spaced equidistantly between 0 and pi) and expiration (bins 4–6 with bins spaced equidistantly between pi and 2pi). At the group level, MEPs were larger in the early inspiration phase (left), with 17/28 participants having their highest respiratory-motor excitability coupling during inspiration (right). **C)** Examples of individual data for cardiac (red) and respiratory (blue) coupling, revealing the extent of intersubject variability. Data supporting panels A–C are available in S2 Data.

coupling during inspiration (participant 22), and lastly an example showing coupling to both rhythms, yet both divergent from the group pattern (participant 26).

This exploratory analysis thus reveals that although group-level effects might indicate alignment to physiological phases of the cardiac and respiratory rhythms, some participants deviate from this alignment yet still display robust coupling with the corticospinal excitability.

### 3.5. No link between self-reported interoception and viscera–motor coupling

We then tested whether the strength of an individual's coupling between visceral phase and corticospinal excitability was related to how much an individual indicates to be aware of this specific organ. To this end, we correlated the coupling strength of each of the organs to the organ-specific scores of the recently developed THISQ questionnaire [41]. There was no significant correlation between these two interoceptive measures (Kendall's tau B correlations; all $p > 0.10$). $BF_{10}$ values indicate moderate evidence for the null hypothesis for the gastric data ($BF_{10} = 0.25$) and an undecided status for the cardiac and respiration data ($BF_{10} = 0.93$ and $BF_{10} = 0.47$, respectively, see Fig 4). Note that for the cardiac and respiratory domains, if anything, the data would point to a negative correlation, whereas one could reasonably assume a priori a positive correlation. Altogether, the results indicate that whether a participant reports high subjective awareness of their heart, lungs, or stomach is unrelated to the extent to which the organ's rhythm is coupled to motor cortex excitability.

Additionally, we tested whether viscera–motor coupling strengths were linked to a commonly used interoception questionnaire (MAIA-2 [42]) that mixes different organs into the same score or sub-score. We specifically focused on the sub-scores most related to awareness of internal sensations, body listening, and noticing. Again, we find no significant correlations with either body listening (cardio $p_{FDR} = 0.20$, $\tau_b = 0.21$, $BF_{10} = 0.75$, respiration $p_{FDR} = 0.12$, $\tau_b = -0.28$, $BF_{10} = 2.04$, gastric $p_{FDR} = 0.98$, $\tau_b = -0.003$, $BF_{10} = 0.24$), or noticing (cardio $p_{FDR} = 0.55$, $\tau_b = -0.08$, $BF_{10} = 0.29$, respiration $p_{FDR} = 0.33$, $\tau_b = -0.22$, $BF_{10} = 0.92$, gastric $p_{FDR} = 0.37$, $\tau_b = -0.16$, $BF_{10} = 0.49$).

Together, these results show that different questionnaires measuring the extent to which someone feels aware of their body, either in a precise organ-specific manner or more generally speaking, are unrelated to the extent to which there is motor-viscera coupling. Thus, self-reports and actual interoceptive coupling with motor excitability tap into different dimensions of interoception.

## 4. Discussion

To quantify organ-specificity in the coupling between interoceptive signals and the motor system, we tested whether the phase of the cardiac, respiratory, and gastric rhythm can explain fluctuations in corticospinal motor excitability. We demonstrate the existence of coupling between motor excitability and the cardiac, respiratory, and gastric rhythms, with similar effect sizes at the group level. However, this apparent similarity did not reflect a consistent pattern across individuals. Rather, participants who showed strong coupling with one rhythm did not necessarily show strong coupling with the others. This pattern of results indicates that the mechanisms underlying the coupling between motor excitability and the three visceral rhythms could be relatively independent from each other. The existence of coupling for all three rhythms despite

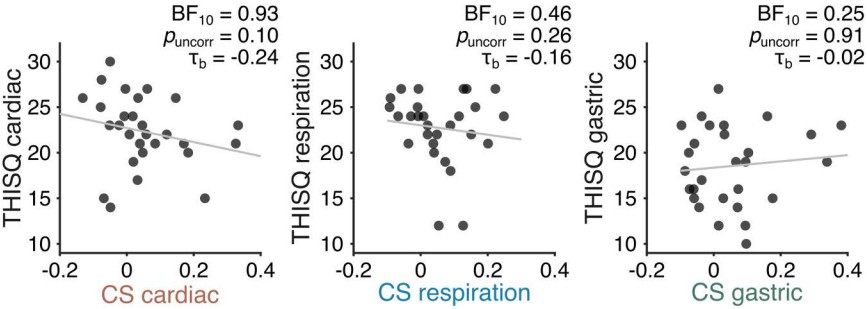

**Fig 4. Absence of significant correlation between self-reported organ-specific interoception on the THISQ questionnaire and the respective coupling strength (CS) with corticospinal excitability of that organ, in the cardiac (left), respiratory (middle), and gastric (right) domains.** Note that for illustration purposes, a regression line has been added to the figure, whereas the reported statistics are the result of a ranked correlation test (Kendall's tau). Notably, in the right panel, the slope of the regression line is positive whereas the tau value is negative.

very different physiological functions and time scales, and the independence of the coupling mechanisms, call for refining and specifying theories offering interpretations on the functional relevance of the impact of visceral rhythms on the brain. Intersubject variability was altogether large: not only participants displayed their own individual interoceptive profiles, but the preferred phase of cardiac or respiratory coupling could also vary. Finally, intersubject variability in viscera–motor coupling was not related to everyday life conscious interoceptive feelings as measured with questionnaires.

### 4.1. Independent influences of the three rhythms suggest at least partially distinct mechanisms

Our results show that motor excitability, measured from the hand region in primary motor cortex down to a specific muscle, is independently related to the cardiac, respiratory, and gastric rhythms. This independence was confirmed in two separate analyses: one showing an absence of correlation between individual coupling strengths between the different organs, and a binarized GLM in which interaction effects between visceral rhythms were likewise absent, with moderate Bayes evidence for absence of interactions in two interactions out of three. Although one previous study revealed effects of cardiac phase on MEP amplitudes [20], null findings have also been reported [21–24]. This discrepancy is likely explained by differences in the experimental design, such as TMS locking to fixed cardiac phases, which might miss the inter-individual variability in optimal phases demonstrated in this experiment, as well as fluctuations in TMS coil to cortical target distance, which was not considered in those studies. To the best of our knowledge, MEP amplitude fluctuations have never been reported along the respiratory or gastric rhythms—and in general, with few notable exceptions such as [16], the literature relating motor activity to interoception considers only one visceral rhythm at a time.

The independence of the effects suggests that each coupling could be mediated by at least partially distinct mechanisms. Our results do not allow to be more specific about the nature of the mechanisms, but it is worth reminding that there are quite a number of possibilities. Firstly, while all three organs are connected to the central nervous system through cranial nerves (vagal and glossopharyngeal) and spino-thalamic pathways (cardiac, phrenic, and splanchnic nerves), some of those pathways might be organ-specific, as recently shown in mice [59,60]. Secondly, each rhythm also has specificities. For instance, cardiac contractions translate into blood pulsations that might be directly transduced in the brain by both neurons and glial cells [61,62], and respiration is detected by olfactory receptors acting also as mechanoreceptors, inducing neural oscillations in large cortical territories [63–66]. Finally, because MEPs reflect a compound measure of cortical, spinal, and muscle excitability, the effects could also partially be driven by peripheral interferences. In cats, motoneuron discharges have been shown to be modulated by both the cardiac cycle [30,67], as well as by respiratory phase [68–70], and arterial pulsations modulate muscle spindle discharge in humans [71]. However, Transcranial Evoked Potentials, i.e., the cortical response to a TMS pulse measured with EEG that reflect cortical excitability only [72], differ in amplitude between cardiac systole and diastole [20]. Therefore, at least the observed modulation of MEPs by the cardiac cycle in the current experiment might represent a mixture of alterations in both cortical and spinal excitability.

Alternatively, cortical motor areas have been shown to control visceral organs, including separate representations of the stomach and diaphragm [73–75] and stimulation of the motor and premotor cortices alters heart rate [76]. The location of these representations with respect to the classical or revised [77] motor homunculus is not yet known. Previous work demonstrated that cortical neurons in rat motor cortex influence specifically sympathetic output to the stomach, and suggested colocalization of skeletomotor and sympathetic function within the same cortical region might facilitate coordination of the two functions [78]. However, here we find the three visceral phase effects on MEP amplitude within the same stimulation site, which might rather indicate global changes in excitability state of the hand motor representation, rather than viscerotopic effects in M1.

Finally, fluctuations in motor cortex excitability might arise from a modulation of the basal ganglia–motor loop. In mice, stimulation of gastric vagal afferents stimulates dopamine release in the nigro-striatal pathway [79]. In humans, deep brain stimulation of the sub-thalamic nucleus in Parkinson patients [80] alleviates not only the motor, but also the gastric symptoms that commonly reported in this pathology [81]. Collectively, those findings suggest that the basal ganglia–motor loop

might be involved in the gastric–motor coupling we observe here—in other words, the coupling itself could begin outside primary motor cortex, but be revealed by TMS stimulation of the primary motor cortex.

Evidence for organ-specific influences on brain signals in humans is still limited. Here, the focus was on excitability changes in the corticospinal motor system, and whether these findings extend to different physiological and cognitive states remains to be tested. Another remaining question is how viscera–viscera interactions moderate effects of visceral rhythms on the brain. Beyond the well-known heart rate modulation by the respiratory rhythm (respiratory sinus arrhythmia), most of the effects of viscera–viscera interactions are still unknown, and might appear as phase–rate coupling, phase–phase coupling, and phase–amplitude coupling. The exact nature of these complex interactions and how they might impact the influence of visceral rhythms on MEPs needs to be addressed in future work.

### 4.2. Functional frameworks should be tightened and refined

We show that all three visceral rhythms independently contribute to motor excitability, which has implications for the functional frameworks that have been proposed to account for the coupling between visceral rhythms and the brain. The active sensing framework [6] is tightly linked to viscera–motor coupling. For example, in rodents respiratory phase is locked to onsets of whisker, nose, and head movements [82,83], which would form a strategic behavioral exploration state [84], and in humans, both visual [8] and tactile exploration [85] are dependent on the cardiac cycle. While the idea that there is a global bodily state, involving both visceral rhythms and motor readiness, favoring exploration is appealing, how could the three visceral rhythms play an independent role in modulating motor excitability in the active sensing framework? Clearly, our data do not allow for a definite answer, and do not dismiss the active sensing framework. Rather, our results call for more specific predictions of the active sensing account, taking into account the existence of independent influences of each visceral rhythm on motor excitability.

Another prominent framework is the scaffolding hypothesis, which proposes that slower bodily rhythms can act as carrier frequencies for neural oscillations, thereby facilitating between-area communication. How could all three visceral rhythms independently fulfill this function? Different timescales of carrier waves could be matched to the temporal structure of incoming information [86], making it advantageous to have various bodily frequencies available. The different visceral rhythms might therefore serve different functions where coupling and decoupling occurs in a state-dependent manner—a proposal for which there is already some experimental evidence: the coupling between the respiratory rhythm and prefrontal cortex is specific to freezing states in mice [87], and there is more coupling of the brain to either the respiratory or the gastric rhythm in rest compared to cognitive tasks, during which temporal structures drastically change [88,89]. As with the active sensing framework, the scaffolding hypothesis needs to be further defined to determine how multiple visceral rhythms could each subserve a function as carrier waves, potentially under different circumstances. Moreover, current frameworks do not yet account for physiological changes occurring at even slower timescales, such as hormonal fluctuations during the menstrual cycle. Although not measured in the current experiment, previous work suggests effects menstrual phase on cortical excitability [90] as well as potential influences on heart–brain interactions [91].

### 4.3. Variety of interoceptive profiles

The involvement of multiple mechanisms of coupling between visceral rhythms and motor excitability, acting with different weights in different participants, could account for the large inter-subject variability observed. Indeed, if at the group level the coupling with motor excitability was similar for all three rhythms, at the individual level all possible combinations of high/moderate/low coupling patterns were observed—for instance, some participants showed large gastric coupling but only marginal respiratory and cardiac coupling, others showed moderate coupling with the three rhythms, etc. In other words, there were as many interoceptive profiles as there were participants. In addition, for the cardiac and respiratory rhythms, the largest motor excitability was most often found during cardiac systole and early inspiration, as would be

expected from the literature [16,20], but a non-negligible proportion of participants displayed equally large coupling during diastole or during expiration. Considering the participant's preferred phase to alter viscero–motor coupling will prove important, both for studies aiming at dissecting mechanisms as well as clinical interventional studies. This sets a limit to the applicability of tasks and interventions in which stimuli are locked to fixed phases of visceral cycles.

Variability in interoceptive profiles was not explained by differences in interoceptive feelings as probed with questionnaire, also known as interoceptive sensitivity [92]. This adds to growing evidence of a gap between objective and subjective measures of interoception [5,93]. For instance, even when directly probed, participants reported no subjective awareness of the objectively measured coupling of motor actions and respiratory phase [16], and the ability at detecting one's own heartbeats is unrelated to interoceptive self-reports [94–96]. Such negative findings may have been partially driven by questionnaires conflating multiple viscera and fundamentally tapping into different constructs [97]. To partly overcome this limitation, here correlations were calculated between organ-specific self-reports [41] and their respective organ–motor coupling score. Despite this, we still found an absence of correlation between self-reported interoception and objectively measured interoceptive influence on motor excitability. It has to be noted that the sample of the current study might not be sufficiently large for the detection of small correlational effects. Whereas the BF indicates some support for the null for the correlation between the THISQ gastric and gastric coupling strength, the Bayes evidence for the other two viscera was inconclusive. Whether and how objective and subjective measures of interoception relate to each other remains an open issue, with potential clinical relevance [3]. Here, future studies would benefit from larger sample sizes and the inclusion of subjective, psychophysical, and neural measures of interoception simultaneously, to further probe the independence of these various levels of interoception. In parallel, more work is needed to fine-tune analysis approaches incorporating phase effects of multiple visceral rhythms at once, and their sensitivity to pick up on the different ways in which these effects could manifest (e.g., binary versus circular effects, as well as different coupling modes [98]).

Lastly, interoception is often suspected to be impaired in psychiatric (see, e.g., [3]) as well as sometimes in neurological disorders (see, e.g., [81])—but in practice in most cases only one aspect of interoception is probed: for instance, cardiac interoception is related to depression but not to anxiety [99,100]. Our results on the variability in interoceptive profiles emphasize the need to test for more than one organ to explore the links between interoception and clinical features.

### 4.4. Conclusion

The coupling between the motor system and gastric, respiratory, and cardiac rhythms can be measured by probing the cortico-spinal excitability of the hand. All three rhythms impact the motor control of the hand, they do so independently, and in different participants the influence of each rhythm is different. These findings have important consequences for how interoception is conceptualized: distinct mechanisms underly coupling between the visceral rhythms and the motor system. Our results call for further specification in frameworks mapping the functional role of coupling between visceral and neural rhythms: why and when does the brain couple to one or multiple of these rhythms and what functions does it serve? Going forward, multi-organ recordings combined with neuroimaging are needed to reveal the functional relevance of individual interoceptive profiles, with potential implications in the fields of psychiatry and neurology.

### Supporting information

**S1 Fig. Individual interoceptive profiles of all 28 participants.** For each participant, the coupling strength between each of the visceral rhythms (red = cardiac, blue = respiratory, green = gastric) and the amplitude of the MEP is plotted. The variability index (v.i.) of each participant reflects how homogeneous or heterogenous coupling strength was across the different organs, with larger values indicating more heterogeneity. Interoceptive profiles show large variability, with both homogenous (e.g., coupling to all, or to none, of the visceral rhythms) and heterogenous (coupling to one or two, but not all visceral rhythms) profiles.
(TIFF)

**S1 Data. Data supporting figure 2 A, B, and D.** Participant mean beta values for distance to target, participant mean coupling strength for each of the visceral rhythms, and coupling strengths for each of the rhythms for the three highest couplers.
(XLSX)

**S2 Data. Data supporting figure 3 panels A–C.** Participant mean phase binned MEP amplitudes corrected for distance for the cardiac and respiratory rhythm.
(XLSX)

## Acknowledgments

The authors thank Camille Straboni for her help in developing the respiration preprocessing pipeline.

## Author contributions

**Conceptualization:** Tahnée Engelen, Teresa Schuhmann, Alexander T. Sack, Catherine Tallon-Baudry.

**Data curation:** Tahnée Engelen.

**Formal analysis:** Tahnée Engelen, Catherine Tallon-Baudry.

**Funding acquisition:** Tahnée Engelen, Catherine Tallon-Baudry.

**Investigation:** Tahnée Engelen.

**Methodology:** Tahnée Engelen, Catherine Tallon-Baudry.

**Project administration:** Tahnée Engelen, Catherine Tallon-Baudry.

**Resources:** Teresa Schuhmann, Alexander T. Sack.

**Supervision:** Catherine Tallon-Baudry.

**Visualization:** Tahnée Engelen.

**Writing – original draft:** Tahnée Engelen, Catherine Tallon-Baudry.

**Writing – review & editing:** Tahnée Engelen, Teresa Schuhmann, Alexander T. Sack, Catherine Tallon-Baudry.

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
