## [Editor Report · Decision Letter 0]

23 Apr 2025

Dear Dr Engelen,

Thank you for submitting your manuscript entitled "The cardiac, respiratory, and gastric rhythms independently modulate motor corticospinal excitability in humans" for consideration as a Research Article by PLOS Biology.

Apologies for the delay in getting back to you, the Easter weekend caused some delays in processing your manuscript.

Your manuscript has now been evaluated by the PLOS Biology editorial staff as well as by an academic editor with relevant expertise and I am writing to let you know that we would like to send your submission out for external peer review.

Once your full submission is complete, your paper will undergo a series of checks in preparation for peer review. After your manuscript has passed the checks it will be sent out for review. To provide the metadata for your submission, please Login to Editorial Manager (https://www.editorialmanager.com/pbiology) within two working days, i.e. by Apr 25 2025 11:59PM.

Kind regards,

Christian

Christian Schnell, PhD

Senior Editor

PLOS Biology

cschnell@plos.org

---

## [Decision Letter · Decision Letter 1]

17 Jun 2025

Dear Dr Engelen,

Thank you for your patience while your manuscript "The cardiac, respiratory, and gastric rhythms independently modulate motor corticospinal excitability in humans" was peer-reviewed at PLOS Biology. It has now been evaluated by the PLOS Biology editors, an Academic Editor with relevant expertise, and by several independent reviewers.

In light of the reviews, which you will find at the end of this email, we would like to invite you to revise the work to thoroughly address the reviewers' reports.

As you will see below, the reviewers find the study very interesting with potentially important insights. Reviewer 1 and Reviewer 2 have a few statistical and methodological concerns that can be addressed through additional analyses and textual revisions. Reviewer 3 additionally wants to see more raw data presented and requests additional analyses.

Given the extent of revision needed, we cannot make a decision about publication until we have seen the revised manuscript and your response to the reviewers' comments. Your revised manuscript is likely to be sent for further evaluation by all or a subset of the reviewers.

**IMPORTANT - SUBMITTING YOUR REVISION**

*Re-submission Checklist*

*Published Peer Review*

*PLOS Data Policy*

*Blot and Gel Data Policy*

Sincerely,

Christian

Christian Schnell, PhD

Senior Editor

PLOS Biology

cschnell@plos.org

REVIEWS:

Reviewer #1 (Micah Allen): This is an insightful and well-executed study that addresses a complex and highly relevant question in neuroscience: how different internal bodily rhythms simultaneously and independently influence brain function, specifically motor corticospinal excitability. The manuscript is clearly written, and the methodologies employed are largely rigorous and appropriate for the research questions. The simultaneous investigation of cardiac, respiratory, and gastric influences on motor evoked potentials (MEPs) represents a novel and important step forward for the field of interoception and brain-body interactions. The findings have significant potential impact, particularly the demonstration of independent modulation by these three rhythms and the concept of individual "interoceptive profiles" that are not captured by self-report measures. The experimental design is careful (e.g., continuous neuronavigation, large trial counts), the statistics are modern and transparent, and the paper stands to unify three organ systems within one protocol, demonstrating their functional independence, which will likely be of broad interest.

The study aimed to determine whether and how the cardiac, respiratory, and gastric visceral rhythms simultaneously and independently modulate corticospinal excitability in humans. The implicit hypothesis was that all three rhythms would show a coupling with MEP amplitude, and that these couplings would be independent of each other, suggesting distinct modulatory pathways. The authors successfully demonstrated that the phase of all three investigated visceral rhythms—cardiac, respiratory, and gastric—significantly modulates MEP amplitude, with similar effect sizes observed at the group level. A key finding was the independence of these modulatory effects: the strength of an individual's viscero-motor coupling for one rhythm did not correlate with their coupling strength for the other two. This independence was further substantiated by a GLM analysis which revealed no significant interaction terms between the influences of the different rhythms. These results strongly suggest that distinct mechanisms underlie the coupling of each visceral rhythm with motor excitability, resulting in unique "interoceptive profiles" for each participant. Additionally, the study found that these objectively measured interoceptive coupling strengths were not correlated with participants' self-reported awareness of their internal organs, as assessed by the THISQ and MAIA-2 questionnaires. Finally, the study also confirmed that TMS coil-to-target distance significantly impacted MEPs, a factor that was appropriately accounted for in the analyses.

The study is of high scientific quality. Its novelty lies in the simultaneous assessment of these three key visceral rhythms and their independent effects on a well-established measure of corticospinal excitability. The use of neuronavigated TMS, comprehensive physiological recordings, and sophisticated statistical analyses (including circular spline GLMs and permutation testing) lends considerable strength to the findings. The paper makes a compelling case for refining current functional frameworks of interoception to account for such independent, multi-organ influences and individual variability.

Major Issues

While the study is robust and the conclusions are generally well-supported, a few areas warrant further attention to enhance its clarity and rigor:

1. The statistical significance of some primary outcomes requires careful consideration regarding multiple comparisons. Specifically, the reported p-values for the coupling strengths of the cardiac and gastric rhythms (p=0.04 for both, from one-sided Wilcoxon signed-rank tests) would not meet conventional significance thresholds if a standard Bonferroni correction for the three primary hypotheses (cardiac, respiratory, gastric) was applied. The manuscript should address this, either by incorporating such an adjustment and discussing its impact on the findings, or by providing a clear rationale for the current statistical approach, perhaps with increased emphasis on the interpretation of the provided Bayes Factor values.

2. The analysis of MEP amplitude involved first residualising it against coil-to-target distance, based on the assumption that variance related to coil distance is purely artifactual. However, as respiratory movements can cause some of the largest coil displacements, this regression could potentially remove a physiological component of interest if breathing-related posture changes also have a direct neural influence pathway. To confirm the robustness of the findings against this assumption, it would be beneficial to discuss this possibility or, ideally, provide a supplementary analysis (e.g., repeating the circular-spline GLM without this regressor, or including it as a linear covariate) to demonstrate that the three rhythm effects persist.

3. The conclusions regarding the independence of coupling strengths among the three rhythms, and the lack of correlation with questionnaire data, are primarily based on non-significant findings from correlation analyses. Given the sample size of N=28, the statistical power to detect small to moderate correlations is limited. A more explicit discussion of this limitation is warranted, for instance by noting that "evidence for absence was modest" where appropriate, particularly when interpreting these null results and considering the Bayes Factors, some of which were inconclusive (e.g., for cardiac and respiratory THISQ correlations). It might also be mentioned in the discussion that future studies should supplement this by utilzing psychophysical measures of objective interoception.

4. The methodology for testing interactions using GLM Model 2 involved binarizing visceral phases into "excitatory" and "inhibitory" periods based on each individual's MEP data. While this approach allows for the inclusion of all three rhythms in a single model, its data-driven nature, which enforces main effects as acknowledged by the authors, and its simplification of continuous phase information, should be more thoroughly discussed. An expanded rationale for choosing this method, along with a consideration of how it might affect sensitivity to detect certain types of interactions, would be beneficial. Furthermore, a brief discussion point acknowledging that future work might explore alternative approaches, such as GLMs incorporating sine and cosine transformed continuous phase angles to model interactions without binarization, could add valuable perspective on methodological advancements.

Minor Issues

- There appears to be a discrepancy in the PDF: the image on page 39 (labelled "Figure 3" in its file) seems to correspond to the text description for Figure 4 (which discusses phase-binned MEPs for cardiac and respiratory rhythms). Conversely, the image on page 40 (labelled "Figure 4" in its file) appears to correspond to the text description for Figure 3 (correlation with THISQ). Please verify and correct.

- The regression of coil-to-target distance is a crucial step. Consider reporting the mean (±SD) coil-to-target distance and perhaps its dominant frequency component or variability to show that residual respiratory or other movement-related artefacts were likely well-controlled by the regression.

- Clarify if the individual stimulation intensity (set to elicit "approximately 1mV") was set relative to a percentage of resting motor threshold (RMT), or if RMT was not formally determined. The group mean ± SD %MSO is reported.

- The manuscript reports the mean (SD) of trials remaining. Could a brief breakdown of the proportion of trials excluded for different reasons (low EMG amplitude, saturation, signal-to-noise ratio, coil displacement) be provided, perhaps in supplementary materials?

- Given that two participants required a relaxed artifact criterion for EGG and the generally lower SNR for EGG, please consider adding a brief note quantifying average cycle variability or another quality metric for the EGG data to allow readers to gauge the reliability of the gastric phase estimates.

- The authors rightly note the gastric rhythm has "no true zero point in the cycle". A brief sentence explaining how the circular spline GLM accommodates this (e.g., by focusing on the modulation shape irrespective of absolute phase origin) could be helpful for some readers.

- Two left-handed participants were included in the final sample of 28. Please briefly justify their inclusion (e.g., stating that their data did not appear as outliers or that repeating analyses without them did not change the main conclusions) or acknowledge this as a minor source of heterogeneity.

- 1000 permutations were used for the null distributions in the circular spline GLM. While the median of the permuted distribution (used for CS calculation) is likely stable, a brief note acknowledging that higher counts (e.g., 10,000) are sometimes advocated for very precise p-value estimation from permutations could be considered if recalculation is not feasible.

- The authors state "custom code and preprocessed data used for the analyses reported in this article are available through osf.io/6utgf/". This is comendable, but do consider if anonymised raw physiological traces could also be uploaded to enhance reproducibility further, if ethically permissible.

- Note in the Methods or Discussion whether female hormonal phase was recorded; if not, briefly acknowledge this as a potential unmeasured variable that can influence corticospinal excitability.

- Typographical Errors:

"matrix sixe" should be "matrix size".

"[z(Coil distance)]" has an unmatched closing bracket in GLM Model 1.

- The discussion generally uses cautious language (e.g., "independently related," "contributed to motor excitability"). Ensure consistency in avoiding overly strong causal claims (e.g., "modulate" implies influence but ensure it's not taken as definitively proven direct causality from this observational design) where correlational or phase-amplitude coupling relationships are described.

- It would be great to see a discussion of these findings in light of recent research on unimodal versus polymodal vagal sensory pathways (e.g., Chang et al., Nature, 2022, https://www.nature.com/articles/s41586-022-04515-5). In particular, considering the differences in methodological scope could be insightful. While invasive animal studies offer greater spatial specificity, the current study provides valuable positive evidence for independent influences in humans. However, the discussion should likely maintain appropriate caution, acknowledging that this single study, focused on corticospinal excitability, may not exhaustively rule out all potential poly-organ or multi-organ interactions that might exist in other physiological or cognitive domains, or be detectable with different methodologies.

Addressing these points will further enhance the clarity, rigor, and impact of this already valuable contribution to the literature on interoception and brain-body dynamics.

Reviewer #2: Overview:

This study aimed to examine how the excitability of the human motor system varies in relation to the phase of three visceral rhythms- cardiac, respiratory, and gastric. To this end, the authors applied single-pulse Transcranial Magnetic Stimulation (TMS) over the hand area of the primary motor cortex to elicit Motor Evoked Potentials (MEPs), while simultaneously recording visceral signals. They then assessed whether MEP amplitude depended on the phase of each visceral rhythm. The authors found that all three rhythms significantly modulated MEP amplitude, indicating a phase-dependent coupling between visceral activity and motor excitability. They also observed that the coupling strengths for each organ were independent of one another - individuals showing strong coupling with one organ did not necessarily show strong coupling with the others. Moreover, these objective coupling measures did not correlate with self-reported interoceptive awareness. The authors concluded that these findings support the existence of distinct, organ-specific mechanisms underlying viscera-motor coupling and highlight the importance of considering individual interoceptive profiles when interpreting brain-body interactions.

Positive features of the study

* Comprehensive, multi-level, and cross-organ investigation: The study examines not only the relationship between MEP amplitude and the phase of organ activity at the group level but also explores inter-individual variability. This approach highlights discrepancies between group-level and individual-level findings and underscores the importance of considering multiple levels of analysis in body-brain interaction research.

* Sophisticated analytical approach: The study employs advanced analytical methods, demonstrating a high degree of methodological rigor and sophistication in addressing the research question.

* Clear writing and engaging figures: The manuscript is well written, with a clear structure and accessible language that effectively conveys complex findings. The figures are visually engaging and thoughtfully designed, further enhancing the clarity of the results.

Major considerations for improvement of the manuscript include the following:

* Need for additional detail in methodological description: The study employs sophisticated data processing and modeling techniques; however, several methodological aspects lack sufficient detail for full comprehension and replication. Providing further information - such as visual examples of raw experimental data, detailed steps for identifying cardiac features (e.g., R-peaks, T-wave offset), and references supporting the computation of coupling strength - would enhance the clarity, accessibility, and reproducibility of the approach.

* Interpreting the functional significance of coupling strength: Although the coupling strengths were statistically greater than zero, their functional relevance remains unclear. While the use of one-sided tests may have contributed to statistical significance, it is still possible that the observed coupling does not reflect a meaningful functional interaction. Contextualizing coupling strength values relative to known benchmarks, if available, would help clarify their significance.

* Caution in interpreting statistical independence: The conclusion that brain-viscera coupling is independent across the three visceral rhythms is based primarily on non-significant findings from generalized linear model (GLM) analyses. However, the absence of statistically significant interaction terms does not conclusively demonstrate functional independence. In my view, the current interpretation may overstate statistical non-significance as evidence of functional independence.

Additional comments:

* Page 3 | Lines 79-81: The statement "Still, no experiment to date measured all three visceral rhythms: the vast majority of experiments are limited to one visceral rhythm, with few notable exceptions combining two (14-16)." may underestimate the number of studies examining cardiac, respiratory, and gastric signals together, especially outside the TMS field. If this limitation is specific to TMS-based studies, it would be helpful to make the scope of the claim more precise.

* Page 7 | Lines 191-192: The description of MEP amplitude as "the difference between the local maximum and local minimum between 20-60 ms after the pulse" would benefit from a reference supporting this methodological choice.

* Page 7 | Line 197: The acronym "RMS" should be defined upon first use.

* Page 7 | Line 201: The method for R-peak detection, using a normalized convolution threshold of 0.6, would benefit from a supporting reference.

* Page 8 | Line 244 - Reference to Liang et al. (2008): This paper primarily addresses variable/model selection, which may not align well with the statistical focus of the current study. A more suitable reference for Bayesian statistics could be Wagenmakers et al. Psychon Bull Rev 2018a, 2018b.

* Page 9 | Line 258: The term "z-scored peak-to-peak amplitude of the MEP" appears to describe the same measure as "difference between the local maximum and local minimum between 20-60 ms after the pulse" in Section 2.7. Consistent terminology across sections would improve clarity.

* Page 9 | Lines 281-282: The sentence "In the null model each data point corresponds to the mean of the data, generating data with a uniform distribution along phase" is unclear. Does "data" refer to MEP amplitude? If so, please clarify.

* Page 9 | Lines 283-284: The definition of the r-value suggests non-negative values. However, Figure 2B displays negative coupling strength values. Please clarify the computation and interpretation of the r-value.

* Page 9 | Line 295: If coupling strength values may have negative values, it may be more appropriate to use a two-sided Wilcoxon signed-rank test rather than a one-sided test against zero.

* Page 11 | Line 338: The phrase "as they were closest to our research questions" is vague. Please elaborate on the rationale for selecting these two MAIA subscales in relation to the research question. Additionally, it would be informative to include correlation results for the other MAIA subscales.

* Page 12 | Line 359: The section title "3.1 Coil distance-to-target explains MEP amplitude" could be rephrased for clarity. Suggested revision: "3.1 Coil-to-target distance explains MEP amplitude."

* Page 12 | Lines 360-361: The sentence "The distance of the TMS coil to the target stimulation site in the motor cortex is known to influence MEP amplitude (40)." may not be best suited for the Results section. To maintain the section's focus on findings, consider moving this contextual information to the Methods section.

* Figure 2D: The visual representation of individuals with the highest coupling strength across organs is unclear. Symbols such as stars and squares may indicate the strongest coupling for one signal but not for others. Please clarify the criteria used to identify these individuals.

* Page 13 | Line 428: The reference to "Figure 4A, left" appears to be a mislabeling and may actually correspond to Figure 3.

* Figure 3: Does "MEP amp | distance" refer to MEP residuals? Also, what do the three boxplots per phase (e.g., systole vs diastole) represent? Please clarify.

* Page 16 | Lines 489-490: The sentence "We found significant coupling between motor excitability and each of the three rhythms, with similar effect sizes at the group level" seems redundant with the preceding sentence. Consider omitting.

* Page 16 | Lines 491-493: The authors present examples of individual variability in coupling across signals. However, it would be valuable to quantify this variability to provide a more robust characterization.

* Page 16 | Line 497: The phrase "specifying the functional interpretative frameworks on interoception" is unclear. Please clarify what is meant by "functional interpretative frameworks" in this context.

* Page 16 | Lines 519-520: The phrase "through cranial nerves and spino-thalamic pathways" could be more specific. If referring to the vagus nerve, it should be explicitly named. Otherwise, clarify which other cranial nerves are involved.

* Page 17 | Lines 520-523: The sentence describes how cardiac and respiratory signals may be sensed in the brain. It would also be relevant to briefly mention how descending (efferent) pathways regulate visceral rhythms (cardiac, respiratory, gastric), how they differ across systems, and how they might interact with descending motor pathways.

* Page 18 | Lines 572-573: The sentence "acting with different weights in different participants, could account for the large inter-subject variability observed" could be supported by a visual representation. Consider adding a spaghetti plot showing individual coupling strengths across systems (e.g., x-axis: organ; y-axis: strength; one line per participant) to illustrate within-subject patterns.

Reviewer #3: Effects of interoception on cortical processing have been scrutinized in many behavioral and neuroimaging experiments. The fundamental question remains: whether these effects can represent changes in the cortical excitability. In the current study the authors addressed this question by investigating effects of three visceral rhythms on the modulation of motor evoked potentials (MEPs) produced by TMS. They found that MEPs were modulated by all three rhythms, yet the modulation by each rhythm was independent of the modulation by other rhythms. The study is performed on a very good technical level and is well presented. I have the following comments:

One should carefully check whether changes in the coupling strength (visceral rhythms vs MEPs) between different rhythms can be due to the different signal-to-noise ratios and thus due to the extractability of the phase portrait which affects the estimation of the coupling strength. This would allow in turn to address the question of whether change in the locking is due to the locking itself or due to a definition of the phase (I assume it is particularly relevant for the gastric activity).

Have authors considered estimating a coupling strength between three visceral rhythms? It might be that the participants with the strongest coupling would have larger co-modulations of MEPs.

It would be desirable to see traces of all three visceral rhythms (are they shown on Fig 1?), particularly gastric activity (over a few cycles). They, of course, can be shown at different times scales.

The authors write: "These results indicate that independent mechanisms underly the coupling between the cardiac, respiratory and gastric rhythm and motor excitability. " Why would we assume that there should be the same mechanism? What would be the reasons?

What is the rationale for the motor cortex being similarly involved across different interceptive systems? What is known about the motor representation for different visceral systems

The authors write: "MEPs were excluded if their amplitude was below 50�V". This potentially can lead to a bias where very small responses are excluded. You can always calculate min-max difference in this interval similar to what you do for pronounced MEPs.

"Their amplitude got cut off due to amplifier saturation" -- for BrainProducts it can only happen if the amplitude of the AD converter resolution during the acquisition was 0.1, with 0.5 it should work quite well without the saturation. What was the setting during the recordings? And how many MEPs were excluded?

Since the amplitude of MEPs depends on the pre-pulse EMG, it would be interesting to see whether correlations across subjects were affected by these pre-stimulus measures.

Can you provide a frequency response for the EGG filter?

---

## [Editor Report · Decision Letter 2]

3 Oct 2025

Dear Dr Engelen,

Thank you for your patience while we considered your revised manuscript "The cardiac, respiratory, and gastric rhythms independently modulate motor corticospinal excitability in humans" for publication as a Research Article at PLOS Biology. This revised version of your manuscript has been evaluated by the PLOS Biology editors and the Academic Editor.

Based on our Academic Editor's assessment of your revision, we are likely to accept this manuscript for publication, provided you satisfactorily address the following data and other policy-related requests:

* We would like to suggest a different title to improve its accessibility for our broad audience:

Cardiac, respiratory and gastric rhythms independently modulate motor corticospinal excitability in humans

* Please add the links to the funding agencies in the Financial Disclosure statement in the manuscript details.

* DATA POLICY:

Regardless of the method selected, please ensure that you provide the individual numerical values that underlie the summary data displayed in the following figure panels as they are essential for readers to assess your analysis and to reproduce it: 2ABD and 3ABC.

* CODE POLICY

We expect to receive your revised manuscript within two weeks.

*Published Peer Review History*

*Press*

Sincerely,

Christian

Christian Schnell, PhD

Senior Editor

cschnell@plos.org

PLOS Biology

---

## [Editor Report · Decision Letter 3]

20 Oct 2025

Dear Dr Engelen,

Thank you for the submission of your revised Research Article "Cardiac, respiratory and gastric rhythms independently modulate motor corticospinal excitability in humans" for publication in PLOS Biology. On behalf of my colleagues and the Academic Editor, Yiheng Tu, I am pleased to say that we can in principle accept your manuscript for publication, provided you address any remaining formatting and reporting issues. These will be detailed in an email you should receive within 2-3 business days from our colleagues in the journal operations team; no action is required from you until then. Please note that we will not be able to formally accept your manuscript and schedule it for publication until you have completed any requested changes.

When you attend to those requests, please also include information in the Methods section of your manuscript whether the study has been conducted according to the principles expressed in the Declaration of Helsinki.

PRESS

Sincerely, 

Christian

Christian Schnell, PhD

Senior Editor

PLOS Biology

cschnell@plos.org